

# Workload assessment for mental arithmetic tasks using the task-evoked pupillary response

Gerhard Marquart and Joost de Winter

Department of BioMechanical Engineering, Faculty of Mechanical, Maritime and Materials Engineering, Delft University of Technology, Delft, The Netherlands

## ABSTRACT

Pupillometry is a promising method for assessing mental workload and could be helpful in the optimization of systems that involve human–computer interaction. The present study focuses on replicating the studies by *Ahern (1978)* and *Klingner (2010)*, which found that for three levels of difficulty of mental multiplications, the more difficult multiplications yielded larger dilations of the pupil. Using a remote eye tracker, our research expands upon these two previous studies by statistically testing for each 1.5 s interval of the calculation period (1) the mean absolute pupil diameter (MPD), (2) the mean pupil diameter change (MPDC) with respect to the pupil diameter during the pre-stimulus accommodation period, and (3) the mean pupil diameter change rate (MPDCR). An additional novelty of our research is that we compared the pupil diameter measures with a self-report measure of workload, the NASA Task Load Index (NASA-TLX), and with the mean blink rate (MBR). The results showed that the findings of Ahern and Klingner were replicated, and that the MPD and MPDC discriminated just as well between the lowest and highest difficulty levels as did the NASA-TLX. The MBR, on the other hand, did not differentiate between the difficulty levels. Moderate to strong correlations were found between the MPDC and the proportion of incorrect responses, indicating that the MPDC was higher for participants with a poorer performance. For practical applications, validity could be improved by combining pupillometry with other physiological techniques.

## INTRODUCTION

Mental workload is an important psychological construct that is challenging to assess on a continuous basis. A commonly used definition of mental workload is the one proposed by *Hart & Staveland (1988)*. These authors defined workload as "the cost incurred by a human operator to achieve a particular level of performance." (p. 140). A valid and reliable assessment method of workload could be helpful in the optimization of systems that involve human–computer interaction, such as vehicles, computers, and simulators. One promising method for measuring workload is pupillometry, which is the measurement of the pupil diameter (e.g., *Goldinger & Papesh, 2012*; *Granholm & Steinhauer, 2004*; *Klingner, Kumar & Hanrahan, 2008*; *Laeng, Sirois & Gredebäck, 2012*; *Marshall, 2007*; *Palinko et al., 2010*; *Schwalm, Keinath & Zimmer, 2008*).

Corresponding author
Joost de Winter,
j.c.f.dewinter@tudelft.nl

Two antagonistic muscles regulate the pupil size: the sphincter and the dilator muscle. Activation of these muscles results in the contraction and dilation of the pupil, respectively. During a mentally demanding task, the pupils have been found to dilate up to 0.5 mm, which is small compared to the maximum dilation of about 6 mm caused by changes in lighting conditions (e.g., *Beatty & Lucero-Wagoner, 2000*). The involuntary reaction of the pupil to changes in task conditions is also called the task-evoked pupillary response (TEPR; *Beatty, 1982*). In the past, TEPRs were obtained at 1–2 Hz by motion picture photography (*Hess & Polt, 1964*). This required researchers to measure the pupil diameter manually frame by frame (*Janisse, 1977*). Nowadays, remote non-obtrusive eye trackers are increasingly being used to automatically measure TEPRs, as these devices are getting more and more accurate.

Over the years, researchers have encountered a few challenges in pupillometry. Reflexes of the pupil to changes in luminance, for example, may undermine the validity of TEPRs. One way to improve validity is to strictly control the luminance of the experimental stimuli, but this limits the usability of pupillometry. *Marshall (2000)* reported she found a way to filter out the pupil light reflex using wavelet transform techniques. She patented this method and dubbed it the "index of cognitive activity". The influence of gaze direction on the measured pupil size is another issue. Where *Pomplun & Sunkara (2003)* reported a systematic dependence of pupil size on gaze direction, *Klingner, Kumar & Hanrahan (2008)* argued that the ellipse-fitting method for the estimation of the pupil size is not affected by perspective distortion.

In the last few decades many researchers have investigated the pupillary response for different types of tasks. Typically, the dilation was found to be higher for more challenging tasks (*Ahern, 1978*; *Kahneman & Beatty, 1966*), including mental arithmetic tasks (*Boersma et al., 1970*; *Bradshaw, 1968*; *Hess & Polt, 1964*; *Schaefer et al., 1968*). Not only task demands have been found to influence the pupil diameter, but also factors like anxiety, stress, and fatigue. *Tryon (1975)* and *Janisse (1977)* extensively reviewed known sources of variation in pupil size. Back then, *Janisse (1977)* commented on the underexplored area of whether pupillary dilations reliably reflect individual differences in intelligence. *Ahern (1978)* discovered that persons scoring higher on intelligence tests showed smaller pupillary dilations on tasks of fixed difficulty. In a more recent study, *Van der Meer et al. (2010)* found greater pupil dilations for individuals with high intelligence than with low intelligence during the execution of geometric analogy tasks. Thus, the results are not consistent and demand further investigation.

The present study focuses on replicating the pupil diameter study by *Ahern (1978)* for mental multiplications of varying levels of difficulty. *Ahern (1978)* found that the more difficult multiplications yielded a greater mean pupil diameter. In her research, *Ahern (1978)* used a so-called television pupillometer (Whittaker 1050S) that was able to measure the pupil diameter in real-time. Specifically, the device processed images obtained from an infrared video camera, identified the pupil diameter using a pattern-recognition algorithm, and computed the diameter of the image of the pupil (*Beatty & Wilson, 1977*). Participants used a chin-rest and infrared eye illuminator, and the camera was positioned

approximately 15 cm from the participant's left eye. Our study is also intended as a follow-up study of *Klingner (2010)*. *Klingner (2010)* recently replicated *Ahern*'s (*1978*) results with a remote eye tracker (Tobii 1750) having a similar working principle as the eye tracker used by *Ahern (1978)*. In *Klingner (2010)*, the participants sat approximately 60 cm from the screen and infrared cameras, and they did not use a chin-rest or head-mounted equipment. In his analyses, *Klingner (2010)* used the average of the two eyes' pupil diameters. With a large number of participants (30 in our study, 39 in *Ahern, 1978*, and 12 in *Klingner, 2010*) and trials (1,350, 1,248, and 632, respectively), and a higher measurement frequency (120 Hz, 20 Hz, and 50 Hz, respectively), the present study aimed to obtain the TEPRs for three levels of difficulty of mental multiplications.

We report the mean pupil diameter change (MPDC) with respect to the baseline pupil diameter right before the presentation of the multiplicand, as was also done by *Ahern (1978)* and *Klingner (2010)*. In addition, we report the absolute mean pupil diameter (MPD). *Laeng, Sirois & Gredebäck (2012)* explained that pupil diameter responses exhibit both a phasic component (i.e., 'rapid' responses to task-relevant events) as well as a tonic component (i.e., 'slow' changes in the baseline pupil diameter). The MPDC allowed us to assess the TEPR, while the MPD allowed us to determine whether the baseline itself differed as a function of the difficulty of the multiplications. Furthermore, in our study, the mean pupil diameter change rate (MPDCR), a measure introduced by *Palinko et al. (2010)*, was examined. The MPDCR is the discrete-time equivalent to the first derivative of the pupil diameter and may be useful for assessing moment-to-moment changes in mental workload. While *Ahern (1978)* and *Klingner (2010)* statistically compared the maximum dilation and mean dilation between the difficulty levels of the mental multiplications, we applied a more fine-grained approach where the MPDC, MPD, and MPDCR were subjected to a statistical test for each 1.5 s time interval in the calculation period. Another way in which our research differs from the works of *Ahern (1978)* and *Klingner (2010)* is that we included two additional measures of mental workload. First, we compared the effect sizes of the pupil diameter measures with those obtained with a classic subjective measurement method of workload, the NASA-TLX. Second, we assessed the mean blink rate (MBR). The relation between mental workload and blink rate has been unclear (*Kramer, 1990*; *Recarte et al., 2008*; *Marquart, Cabrall & De Winter, 2015*), and our aim was to clarify this relationship.

The numbers in our study were presented visually in order to gain temporal consistency, as was also done by *Klingner* (*2010*; cf. *Ahern, 1978*, in which the numbers were presented aurally). Furthermore, as in *Klingner (2010)*, the pupil diameter was recorded with an automatic remote eye tracker (SmartEye DR120).

## METHOD

### Ethics statement

The research was approved by the Human Research Ethics Committee (HREC) of the Delft University of Technology (TU Delft 'Workload Assessment for Mental Arithmetic Tasks

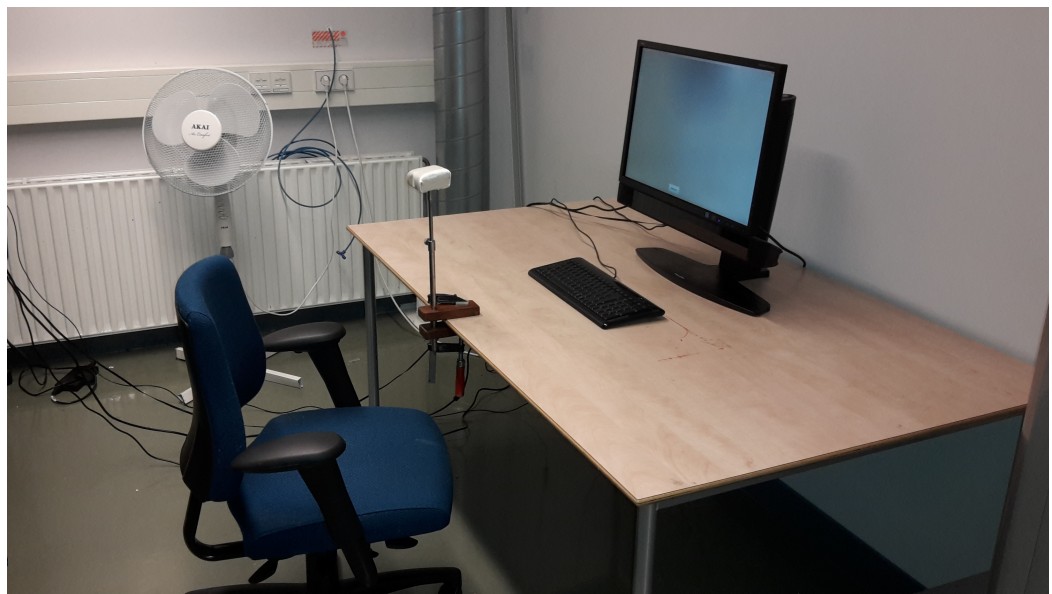

**Figure 1** Experimental equipment: monitor with built-in eye tracker (SmartEye DR120), chin-rest, and keyboard.

using the Task-Evoked Pupillary Response: January 29, 2015). All participants provided written informed consent.

## Participants

Thirty participants (2 women and 28 men), aged between 19 and 38 years ($M = 23$, $SD = 4.1$ years) were recruited to volunteer in this experiment (25 BSc/MSc students and 5 persons with an MSc degree). Individuals wearing glasses or lenses were excluded from participation. All participants read and signed an informed consent form, explaining the purpose and procedures of the experiment and received € 5 compensation for their time.

## Equipment

The SmartEye DR120 remote eye tracker, with a sampling rate of 120 Hz, was used to record the participant's pupil diameter, eyelid opening, and gaze direction while sitting behind a desktop computer (see Fig. 1). The pupil diameter was the average of the left and right pupil diameter, as provided by the SmartEye 6.0 software. The software estimates the pupil diameter as the major axis of an ellipse that is fit to the edge of the pupil. In order to obtain more accurate measurements, a chin-rest was used. The eye tracker was equipped with a 24-inch screen, which was positioned approximately 65 cm in front of the sitting participant and which was used to display task-relevant information. The outcome of a task had to be entered using the numeric keypad of a keyboard (cf. *Ahern, 1978* in which participants used a keyboard, and *Klingner, 2010* in which participants used a touchscreen).

The experiment took place in a room where there was office lighting delivered by standard fluorescent lamps and where daylight could not enter. Our approach to room illumination was similar to that used by *Klingner (2010)*. We acknowledge that a stricter

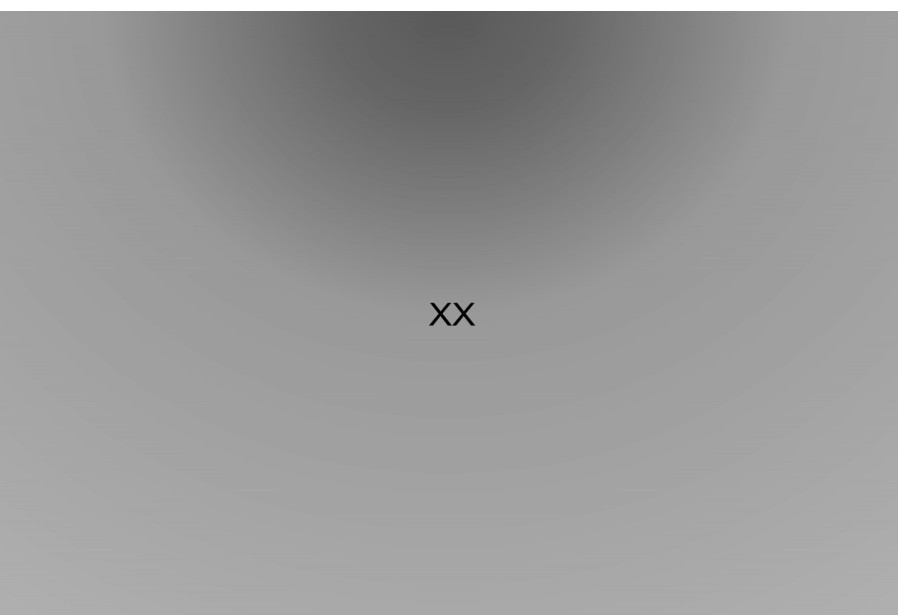

**Figure 2  Task display during accommodation, pause, and calculation period.**

control of lighting is possible. For example, *Janisse (1977)* reported that he ensured constant illumination of his experimental lab by feeding all electric current used in the room through a constant voltage transformer. No such strict control of illumination was applied in our research nor did we measure the degree of ambient lighting. However, because the experimental conditions were counterbalanced, we reasoned that there could be no systematic effect of ambient lighting on our results. Furthermore, we used a screen background with variable brightness, designed to minimize the pupillary light reflex in case a participant looked away from the center of the screen (Fig. 2; *Marquart, 2015*). The corresponding image file is available in Supplemental Information.

## Procedure

The participants were requested to perform 50 trials of mental arithmetic tasks (multiplications of two numbers), five of which were used as a short training. The remaining 45 trials were presented in three sessions of different levels of difficulty (easy, medium, and hard; see Table S1). Level 1 contained the 15 easiest multiplications (outcomes ranging between 80 and 108), Level 2 contained 15 multiplications of intermediate difficulty (outcomes between 126 and 192), and Level 3 contained the 15 hardest multiplications (outcomes between 221 and 324).

The sequence of the three sessions was counterbalanced across the participants. Each trial was initiated by the participant by pressing the enter key and started with a 4 s accommodation period, followed by a 1 s visual presentation of two numbers (multiplicand and multiplier) between 6 and 18, with a 1.5 s pause in between (Table 1). The participants were asked to multiply the two numbers and type their answer on the numeric keypad 10 s after the multiplier disappeared. Thus, the total duration of one trial

**Table 1** Timeline of an individual trial.

| Period | Start time (s) | End time (s) | Symbol |
|---|---|---|---|
| Accommodation | 0.0 | 4.0 | XX |
| Baseline | 3.6 | 4.0 | XX |
| Multiplicand | 4.0 | 5.0 | 08 |
| Pause | 5.0 | 6.5 | XX |
| Multiplier | 6.5 | 7.5 | 16 |
| Calculation | 7.5 | 17.5 | XX |
| Response | 17.5 | When pressing enter key | N/A |

was 17.5 s ($4 + 1 + 1.5 + 1 + 10$). When the numbers were not presented, a double "X" was shown to avoid pupillary reflexes caused by changes in brightness or contrast.

After each of the three sessions, participants were asked to fill out a NASA-TLX questionnaire to assess their subjective workload on six facets: mental demand, physical demand, temporal demand, performance, effort, and frustration (*Hart & Staveland, 1988*). All questions were answered on a scale from 0% (very low) to 100% (very high). For the performance question, 0% meant perfect and 100% was failure. The participants' overall subjective workload was obtained by averaging the scores across the six items. The total duration of the experiment was approximately 30 min.

## Instructions to participants

Before the experiment started, the participants were informed that they had to do 50 multiplications, five of which would be used as a short training. They were also told that the remaining 45 trials were presented in three sessions of varying difficulty (easy, medium, and hard). The participants were requested to position themselves in front of the monitor with their chin leaning on the chin-rest. They were instructed to stay still, keep their gaze fixed, focus (not stare) at the center of the screen throughout a trial. In addition, participants were asked to blink as little as possible, obviously without causing irritation, and to start each trial with 'a clear mind' (i.e., not thinking about the previous trial). If the participants could not complete the multiplication, they were instructed to enter zero as their answer.

## Data processing

The data were processed in two steps. In the first step, the missing values in the pupil diameter data (lost during recording) were removed and the signals were repaired with linear interpolation (see Fig. 3A, for an illustration). On average, 1.2% of the data were lost, so this processing step did not substantially influence the results. In the second step, blinks and poor-quality data were removed. During a blink, the eyelid opening rapidly diminishes and then increases in a few tenths of a second until it is fully open again. It is impossible to track the pupil diameter while blinking. The pupil diameter quality signal (provided by the SmartEye software) was used to filter out the poor quality data. This signal ranges from 0 to 1, with values close to 1 indicating a good quality (*SmartEye, 2013*). All data points with a pupil diameter quality below 0.75 were removed. Trials containing less than 70% of the data were excluded from the analysis. Of the initial 1,350 trials from 30 participants, 1,125

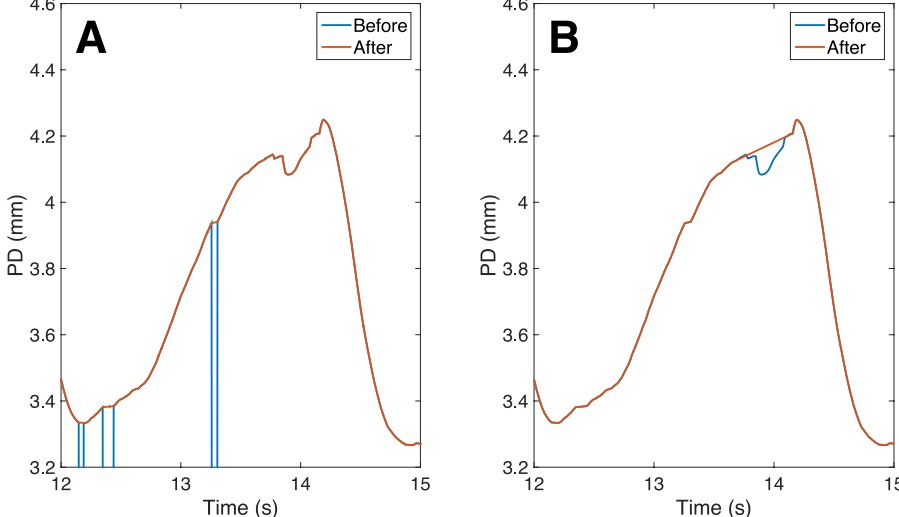

**Figure 3** **Illustration of data processing.** (A) Pupil diameter (PD) before and after linear interpolation for missing values. (B) Pupil diameter before and after linear interpolation for poor-quality data.

trials passed these criteria (394 for Level 1, 384 for Level 2, & 347 for Level 3; the entire level 2 session of one participant [15 trials] was discarded). The gaps in the 1,125 trials were filled using linear interpolation (Fig. 3B).

The last 0.4 s of the accommodation period was defined as the pupillary baseline, as was done by *Klingner (2010)*. The mean pupil diameter of the baseline period (3.6–4.0 s) of each trial was subtracted from each trial to accommodate for any possible shifts or drifts. The mean pupil diameter change (MPDC) for each participant was then obtained by averaging all trials per level of difficulty. Similarly, the mean pupil diameter (MPD) for each participant was obtained but then without subtracting the mean pupil diameter of the baseline period. The MPDCR was calculated for each participant as the average velocity (mm/s) or change in MPD between two points in time. In order to compare the three difficulty levels, the MPD and MPDC were analyzed at eight fixed points in time from the multiplier and calculation periods (i.e., P1 = 6.5 s, P2 = 7.5 s, P3 = 9.0 s, P4 = 10.5 s, P5 = 12.0 s, P6 = 13.5 s, P7 = 15.0 s, P8 = 16.5 s). The MPDCR was assessed across the seven interim periods.

In addition to these analyses, the mean blink rate (MBR) for two different periods in time was calculated. That is, a distinction was made between low mental demands (i.e., from the beginning of the accommodation period until the presentation of the multiplier; i.e., from 0 to 6.5 s) and high mental demands (i.e., from the presentation of the multiplier until the end of the calculation period; i.e., from 6.5 to 17.5 s). A blink was defined as the moment that the eye opening dropped below 75% of the mean eyelid opening of that trial (see Fig. S1).

## Statistical analyses

The pupil diameter measures (MPD, MPDC, and MPDCR), the blink rates (MBR), and the results of the NASA-TLX were analyzed with paired *t*-tests between the three levels

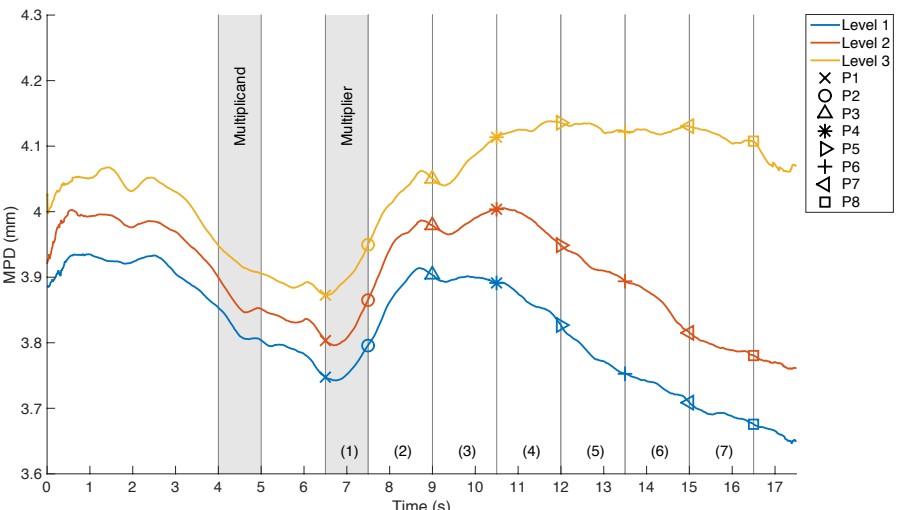

**Figure 4 Mean pupil diameter (MPD) during the mental multiplication task for the three levels of difficulty.** The grey bars represent the periods where the multiplicand and multiplier were shown on the screen. The numbers were masked by an "XX" during the remainder of the trial.

(i.e., Level 2 vs. 1, Level 3 vs. 1, and Level 3 vs. 2). Additionally, Pearson's $r$ correlation coefficients were obtained between the MPDC, the NASA-TLX, and the percentage of incorrect responses. For all analyses, a Bonferroni correction was applied. Accordingly, we set the significance level to 0.05/3 ($\sim$0.0167).

Cohen's $d_z$ effect size (see Eq. (1)) was calculated to determine at which points in time the differences in MPDC between the three levels of difficulty were largest. In Eq. (1), $M$ and $SD$ are the mean and standard deviation of the vector of data points, respectively, $r$ is the Pearson correlation coefficient between the two vectors of data points, $t$ is the $t$-statistic of a paired $t$-test, and $N$ is the sample size (i.e., the number of pairs, which was either 29 or 30).

$$d_z = \frac{M_i - M_j}{\sqrt{SD_i^2 + SD_j^2 - 2 * r * SD_i * SD_j}} = \frac{t}{\sqrt{N}}. \tag{1}$$

## RESULTS

### Mean pupil diameter (MPD)

The MPD during the mental multiplication task is shown in Fig. 4. It can be seen that at all points in time, the MPD was higher for the higher levels of difficulty. The pattern of the MPD was similar for all levels during the first ten seconds. Figure also shows the results for the period 6.5–17.5 s, split into seven periods with eight points. The means and standard deviations of the MPD for the eight points in time and the three levels of difficulty are shown in Table 2, together with the effect sizes ($d_z$) and the $p$-values of the pairwise comparisons. The results confirm that the MPD was significantly higher for the more difficult levels at all points in time.

**Table 2 Mean pupil diameter (MPD), mean pupil diameter change (MPDC), mean pupil diameter change rate (MPDCR), NASA-TLX, and mean blink rate (MBR), per level of difficulty of the multiplications.** The means ($M$) and standard deviations ($SD$) are shown per level of difficulty of the multiplications. P1–P8 refers to the eight points in time, while (1)–(7) refers to the seven periods. Statistically significant differences are indicated in boldface. $N = 30$ for the NASA-TLX for all three levels.

| | M(SD) | | | p-value ($d_z$) | | |
|---|---|---|---|---|---|---|
| | Level 1 (N = 30) | Level 2 (N = 29) | Level 3 (N = 30) | Level 2 vs. 1 (df = 28) | Level 3 vs. 1 (df = 29) | Level 3 vs. 2 (df = 28) |
| **MPD (mm)** | | | | | | |
| P1 | 3.748 (0.456) | 3.804 (0.467) | 3.873 (0.490) | 0.334 (0.18) | **0.001** (0.71) | 0.026 (0.44) |
| P2 | 3.796 (0.480) | 3.865 (0.486) | 3.949 (0.516) | 0.119 (0.30) | **<0.001** (0.84) | **0.009 (0.53)** |
| P3 | 3.904 (0.470) | 3.979 (0.481) | 4.051 (0.531) | 0.107 (0.31) | **<0.001** (0.79) | 0.036 (0.41) |
| P4 | 3.891 (0.456) | 4.003 (0.478) | 4.113 (0.522) | 0.037 (0.41) | **<0.001** (1.04) | **0.007 (0.54)** |
| P5 | 3.827 (0.429) | 3.948 (0.488) | 4.136 (0.521) | 0.017 (0.47) | **<0.001** (1.47) | **<0.001 (0.84)** |
| P6 | 3.752 (0.451) | 3.894 (0.490) | 4.122 (0.518) | 0.017 (0.47) | **<0.001** (1.57) | **<0.001 (0.88)** |
| P7 | 3.709 (0.427) | 3.815 (0.474) | 4.130 (0.500) | 0.051 (0.38) | **<0.001** (1.73) | **<0.001 (1.26)** |
| P8 | 3.676 (0.436) | 3.781 (0.460) | 4.108 (0.493) | 0.064 (0.36) | **<0.001** (1.94) | **<0.001 (1.21)** |
| **MPDC (mm)** | | | | | | |
| P1 | −0.118 (0.087) | −0.114 (0.115) | −0.093 (0.085) | 0.837 (0.04) | 0.158 (0.26) | 0.424 (0.15) |
| P2 | −0.069 (0.094) | −0.052 (0.118) | −0.017 (0.120) | 0.310 (0.19) | **0.016** (0.47) | 0.218 (0.23) |
| P3 | 0.038 (0.148) | 0.061 (0.148) | 0.084 (0.152) | 0.297 (0.20) | 0.107 (0.30) | 0.452 (0.14) |
| P4 | 0.026 (0.179) | 0.086 (0.149) | 0.147 (0.171) | 0.039 (0.40) | **0.001** (0.65) | 0.093 (0.32) |
| P5 | −0.038 (0.204) | 0.031 (0.164) | 0.169 (0.205) | **0.013** (0.49) | **<0.001** (1.13) | **<0.001** (0.74) |
| P6 | −0.113 (0.196) | −0.024 (0.193) | 0.155 (0.228) | **0.012** (0.50) | **<0.001** (1.50) | **<0.001** (0.86) |
| P7 | −0.156 (0.186) | −0.102 (0.207) | 0.164 (0.226) | 0.044 (0.39) | **<0.001** (1.94) | **<0.001** (1.35) |
| P8 | −0.190 (0.179) | −0.136 (0.208) | 0.143 (0.248) | 0.115 (0.30) | **<0.001** (1.95) | **<0.001** (1.20) |
| **MPDCR (mm/s)** | | | | | | |
| (1) | 0.048 (0.087) | 0.062 (0.079) | 0.076 (0.112) | 0.210 (0.24) | 0.068 (0.35) | 0.463 (0.14) |
| (2) | 0.072 (0.080) | 0.076 (0.069) | 0.067 (0.081) | 0.696 (0.07) | 0.765 (−0.06) | 0.698 (−0.07) |
| (3) | −0.008 (0.078) | 0.016 (0.070) | 0.042 (0.055) | 0.094 (0.32) | **0.002** (0.61) | 0.088 (0.33) |
| (4) | −0.043 (0.052) | −0.037 (0.057) | 0.015 (0.052) | 0.606 (0.10) | **<0.001** (0.99) | **<0.001** (0.74) |
| (5) | −0.050 (0.060) | −0.036 (0.059) | −0.009 (0.067) | 0.514 (0.12) | 0.021 (0.45) | 0.052 (0.38) |
| (6) | −0.029 (0.051) | −0.053 (0.053) | 0.006 (0.060) | 0.098 (−0.32) | **0.015** (0.47) | **<0.001** (0.78) |
| (7) | −0.022 (0.052) | −0.022 (0.062) | −0.014 (0.051) | 0.827 (−0.04) | 0.514 (0.12) | 0.372 (0.17) |
| **NASA-TLX (%)** | | | | | | |
| Total | 21 (13) | 31 (13) | 49 (14) | **<0.001** (0.86) | **<0.001** (1.91) | **<0.001** (1.48) |
| Mental | 34 (21) | 47 (17) | 70 (17) | **0.002** (0.63) | **<0.001** (1.39) | **<0.001** (1.51) |
| Physical | 16 (17) | 19 (19) | 20 (20) | 0.045 (0.38) | 0.118 (0.29) | 0.707 (0.07) |
| Temporal | 19 (15) | 29 (18) | 53 (23) | **0.004** (0.56) | **<0.001** (1.41) | **<0.001** (1.26) |
| Performance | 10 (12) | 21 (17) | 40 (23) | **0.002** (0.62) | **<0.001** (1.45) | **<0.001** (0.91) |
| Effort | 28 (19) | 43 (17) | 64 (22) | **<0.001** (0.75) | **<0.001** (1.35) | **<0.001** (1.15) |
| Frustration | 18 (17) | 27 (24) | 45 (29) | **0.005** (0.56) | **<0.001** (1.21) | **<0.001** (0.85) |
| **MBR (blinks/s)** | | | | | | |
| (0.0–6.5 s) | 0.262 (0.165) | 0.258 (0.168) | 0.303 (0.216) | 0.748 (0.06) | 0.203 (0.24) | 0.265 (0.21) |
| (6.5–17.5 s) | 0.218 (0.187) | 0.212 (0.175) | 0.265 (0.210) | 0.861 (0.03) | 0.078 (0.33) | 0.023 (0.44) |

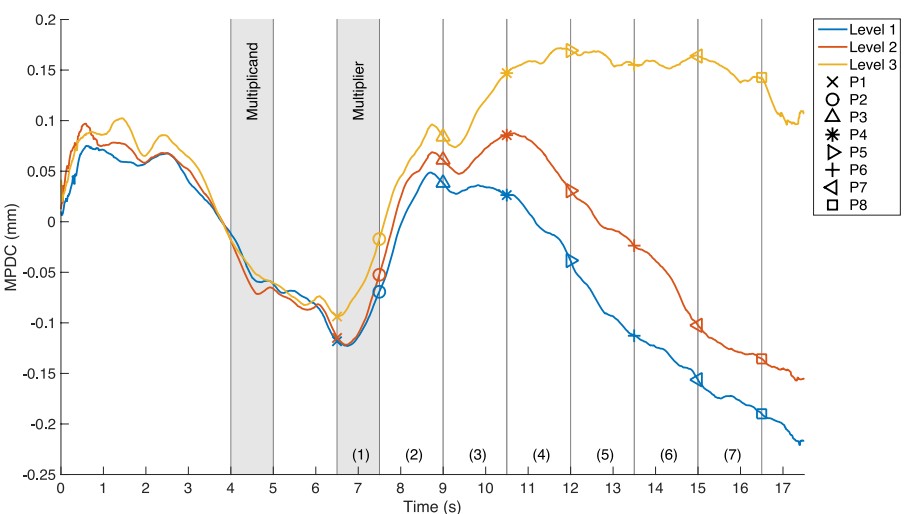

**Figure 5** **Mean pupil diameter change (MPDC) during the mental multiplication task, for the three levels of difficulty.** The grey bars represent the periods where the multiplicand and multiplier were shown on the screen. The numbers were masked by an "XX" during the remainder of the trial.

## Mean pupil diameter change (MPDC)

Figure 5 shows the MPDC as a function of the level of difficulty. As mentioned above, this measure takes into account the shift of the baseline by subtracting the mean of the baseline period of each trial. The difference between the three pupillary responses during the calculation period can now be seen more clearly as compared to the MPD. Again, the multiplier and calculation periods were split into seven periods by eight points. The results of the analysis of the MPDC at the eight points in time and the three levels of difficulty are shown in Table 2. A significant difference occurred at Points 4–8. The effect size estimate Cohen's $d_z$ was also calculated for the MPDC between pairs of difficulty levels for each point in time (see Fig. 6). It can be seen that large effect sizes arose after approximately 11 s since the start of the trial, especially between Levels 1 and 3.

## Mean pupil diameter change rate (MPDCR)

Figure 7 shows the MPDCR as a function of the difficulty level for the seven periods. A positive value indicates overall pupil dilation during that period and a negative value means overall contraction of the pupil diameter. In the first two periods, the diameter increased with approximately equal velocity for the three levels. During the other periods, the velocities decreased and became negative. Significant differences were found between the three conditions (see also Table 2).

## Self-reported workload (NASA-TLX)

The results of the NASA-TLX questionnaire are shown in Fig. 8. For almost all items, the TLX score was significantly higher for the more difficult multiplications (see also Table 2). Only the subjective physical workload did not differ significantly between the levels of difficulty.

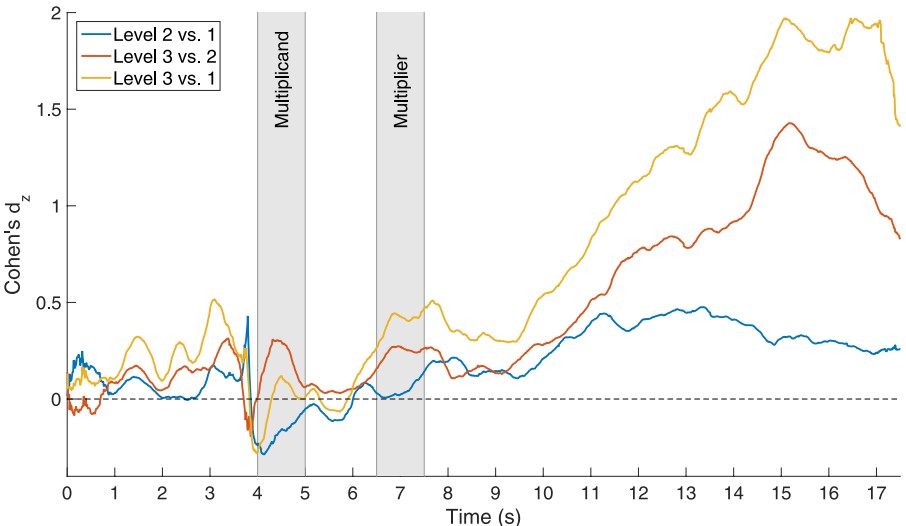

**Figure 6 Cohen's $d_z$ for the mean pupil diameter change (MPDC) between pairs of levels of difficulty.** The grey bars represent the periods where the multiplicand and multiplier were shown on the screen. The numbers were masked by an "XX" during the remainder of the trial.

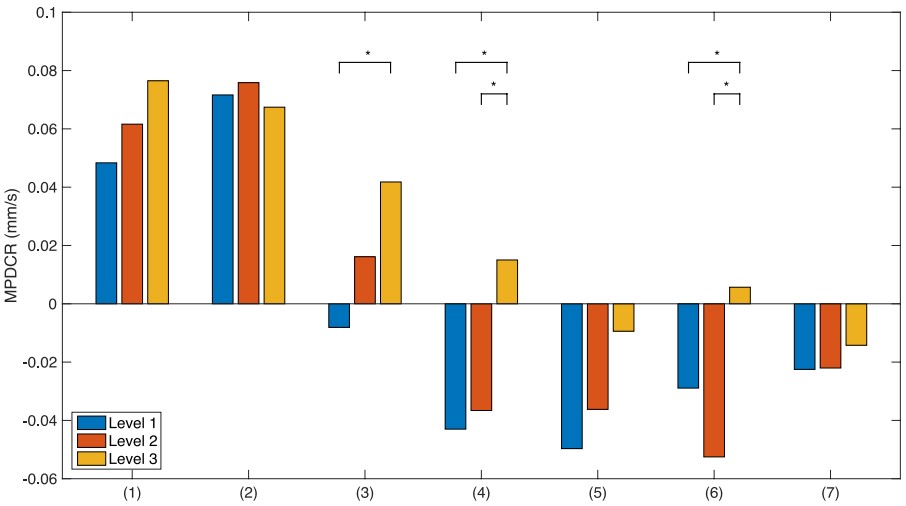

**Figure 7 Mean pupil diameter change rate (MPDCR), for the three levels of difficulty and for seven periods in time during the presentation of the multiplier and the calculation period.** The asterisks indicate statistically significant differences between the levels of difficulty.

## Pupil diameter of correct versus incorrect responses

The percentages of correct responses for Levels 1, 2, and 3 were respectively 94.7%, 92.9%, and 66.4% when selecting all 450 trials per level. When considering only those trials which passed the data filtering (see section 'Data processing'), the percentages of correct responses for Levels 1, 2, and 3 were respectively 94.2% (371 of 394 trials), 93.8% (360 of 384 trials), and 69.2% (240 of 347 trials). Figure 9 shows the MPD for Level 3 separated into correct and incorrect responses. Too few incorrect answers were given for

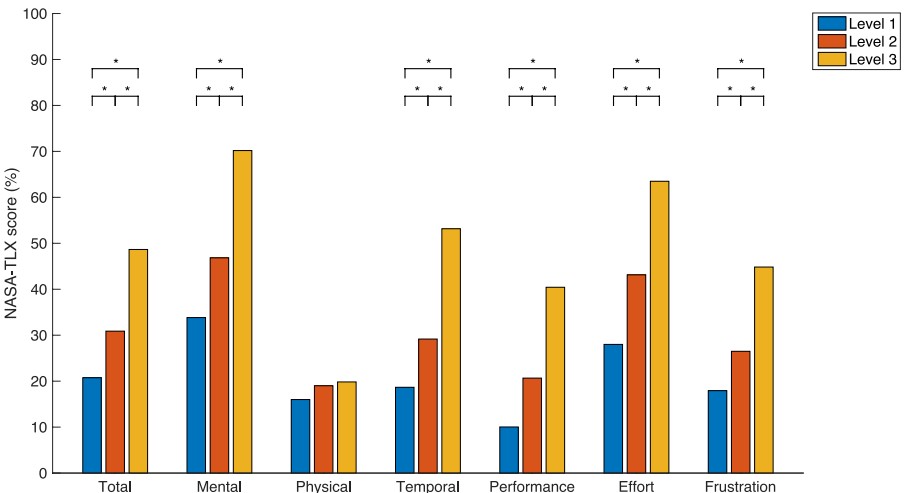

**Figure 8** **Results of the NASA-TLX questionnaire, for the three levels of difficulty.** The asterisks indicate statistically significant differences between the levels of difficulty.

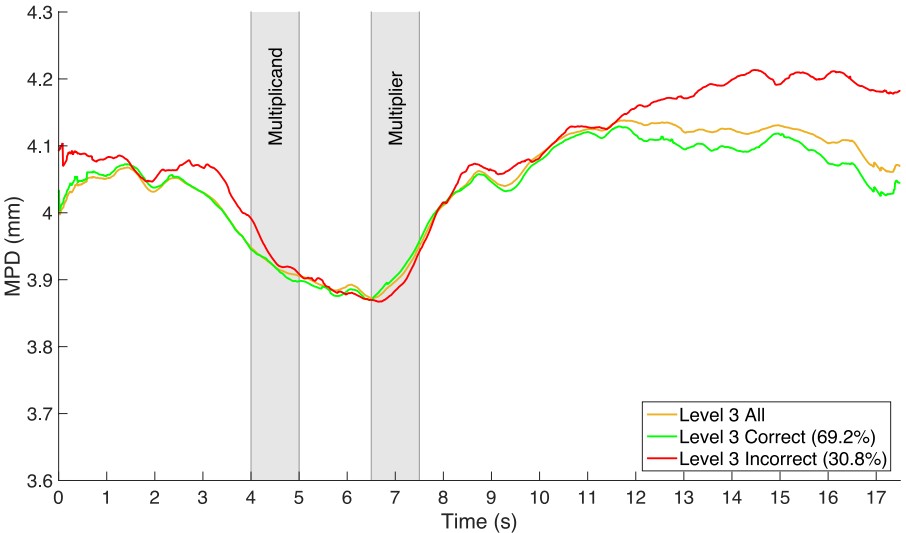

**Figure 9** **Mean pupil diameter (MPD) during the mental multiplication task for the third level of difficulty. A distinction is made between correct and incorrect responses.** The grey bars represent the periods where the multiplicand and multiplier were shown on the screen. The numbers were masked by an "XX" during the remainder of the trial.

the other two levels and the results for these levels are therefore not reported. There were no significant differences between the MPD for correct and incorrect responses (Table S2).

## Blink rate

Table 2 shows that the MBR of Level 3 was higher, but not significantly so, than the MBR of Levels 1 and 2. However, for each level of difficulty, the MBR was higher during periods with low mental demands (0–6.5 s) than during higher mental demands (6.5–17.5 s). Figure 10 illustrates the cumulative number of blinks as a function of time. It can be seen

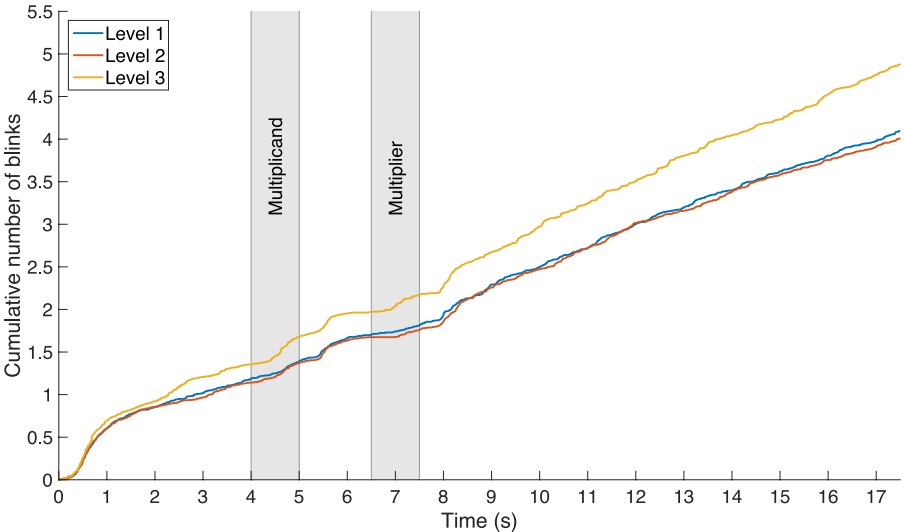

**Figure 10  Mean cumulative number of blinks during the mental multiplication task for the three levels of difficulty.**

that participants were likely to blink at distinct moments in time, namely right after the start of the trial ($\sim$0.5 s), right after the presentation of the multiplicand ($\sim$4.5 s), and after the presentation of the multiplier ($\sim$8.0 s).

## Correlations between MPDC, NASA-TLX, and proportion of incorrect responses

The results of the correlation analyses between the MPDC, NASA-TLX, and proportion of incorrect responses are shown in Table 3. For the MPDC and NASA-TLX, the table shows overall positive correlations, for the eight points in time and for the three different levels of difficulty. Between the MPDC and the percentage of incorrect responses, three statistically significant positive correlation coefficients were observed at Points 1 and 2. Furthermore, Table 3 shows that people who experienced higher subjective workload (i.e., a higher NASA-TLX score) generally gave more incorrect responses.

## DISCUSSION

### Pupil diameter results

The results showed that the MPD was higher for the higher levels of difficulty at all eight points of the calculation period, with Points 7 and 8 exhibiting the largest differences. The MPD findings demonstrate that the baseline of the pupil diameter can shift during mental activity. If the pupil had been given more time to recover from the previous trial by increasing the length of the accommodation period, the difference of the MPD between the three levels of difficulty in the first period would probably have been smaller.

A remarkable finding is the behavior of the MPD during the first 2.5 s of the accommodation period. Where a clear decline from the start or a horizontal line might be expected, the MPD starts to decline only after about 2.5 s. This unexpected finding may have been caused by the fact that participants looked away from the center of the

**Table 3** Pearson's correlations ($r$) between the mean pupil diameter change (MPDC), percentage of incorrect responses, and the overall NASA-TLX scores, for the three levels of difficulty. Statistically significant correlations are indicated in boldface.

|  | Level 1 | Level 2 | Level 3 | Mean of levels 1–3 |
|---|---|---|---|---|
|  | $r$ ($p$-value) | $r$ ($p$-value) | $r$ ($p$-value) | $r$ ($p$-value) |
| **MPDC vs. overall NASA-TLX** | | | | |
| P1 | −0.02 (0.899) | 0.20 (0.310) | 0.20 (0.283) | 0.33 (0.072) |
| P2 | −0.22 (0.239) | 0.29 (0.130) | 0.09 (0.644) | 0.17 (0.376) |
| P3 | −0.15 (0.523) | 0.04 (0.818) | 0.01 (0.978) | 0.01 (0.965) |
| P4 | 0.09 (0.435) | 0.07 (0.733) | 0.04 (0.833) | 0.17 (0.365) |
| P5 | 0.11 (0.641) | 0.11 (0.554) | 0.02 (0.925) | 0.09 (0.654) |
| P6 | 0.05 (0.550) | 0.20 (0.307) | −0.01 (0.952) | 0.09 (0.637) |
| P7 | 0.05 (0.813) | 0.20 (0.290) | 0.17 (0.363) | 0.14 (0.469) |
| P8 | −0.00 (0.998) | 0.26 (0.176) | 0.16 (0.385) | 0.18 (0.349) |
| **MPDC vs. % incorrect responses** | | | | |
| P1 | 0.34 (0.063) | 0.44 (0.017) | 0.35 (0.061) | 0.64 (**<0.001**) |
| P2 | 0.17 (0.371) | 0.51 (**0.005**) | 0.30 (0.110) | 0.59 (**0.001**) |
| P3 | 0.03 (0.882) | 0.26 (0.180) | 0.11 (0.567) | 0.22 (0.244) |
| P4 | 0.23 (0.219) | 0.25 (0.183) | 0.16 (0.385) | 0.36 (0.051) |
| P5 | 0.16 (0.397) | 0.16 (0.409) | 0.06 (0.749) | 0.25 (0.179) |
| P6 | 0.03 (0.882) | 0.21 (0.285) | 0.04 (0.847) | 0.16 (0.396) |
| P7 | −0.00 (0.995) | 0.32 (0.090) | 0.14 (0.459) | 0.28 (0.137) |
| P8 | 0.04 (0.838) | 0.25 (0.193) | 0.14 (0.454) | 0.24 (0.197) |
| **Overall NASA-TLX vs. % incorrect responses** | | | | |
|  | 0.57 (**0.001**) | 0.35 (0.056) | 0.53 (**0.002**) | 0.58 (**<0.001**) |

screen when their outcome to the multiplication had to be entered. Although the responses were not given during the accommodation period, the fluctuation could be an aftereffect because the trials came in relatively quick succession. During the presentation of the multiplicand and the pause (4–6.5 s) the MPD decreased further, at a slower pace however, which seems to indicate memory load (cf. *Kahneman & Beatty, 1966*). A small increase of the pupil diameter after the presentation of the first number was observed by *Ahern (1978)* and *Klingner (2010)*.

The MPDC has the advantage compared to MPD that it corrects for fluctuations in the baseline pupil diameter, and hence compensates for any structural temporal trends that might exist. The use of MPDC is appropriate as compared to other types of measures such as percent dilation, because as pointed out by *Beatty & Lucero-Wagoner (2000)*, "the extent of the pupillary dilation evoked by cognitive processing is independent of baseline pupillary diameter over a wide range of baseline values." (p. 148). What is notable in the MPDC results (Fig. 5) is that the pupillary behavior between the three difficulty levels was highly similar during the first few seconds after the presentation of the multiplier (6.5–9 s). This might be due to the strategy that the participants used. One can imagine that the first step in each multiplication, regardless of its difficulty, is similar. For example, the

first step for many people of the Level 1 multiplication 7 × 14 would probably be 7 × 10. This is comparable to the first step of the Level 3 multiplication 14 × 18, which would then be 14 × 10. These observations are in line with the TEPRs obtained by *Ahern (1978)*, who found a similar response between the three levels of difficulty at the beginning of the calculation. The MPDC during the other periods was found to differ significantly between the three levels, particularly when Levels 1 and 2 were compared to Level 3.

The results of the MPDCR illustrate that the effect sizes are smaller when compared to the results of the MPDC measure. Presumably, the MPDCR is less sensitive to changes in mental workload because it represents second-to-second changes in pupil diameter rather than the actual pupil diameter itself (either absolutely as in the MPD, or relative to a baseline as in the MPDC). As with any first-order derivative of a signal, the MPDCR might be more sensitive to noise and unsystematic moment-to-moment fluctuations in pupil diameter. Nonetheless, the MPDCR does provide a clear indication of when the muscles of the pupil respond, and hence when the mental workload increases or decreases.

An interesting question related to Fig. 9 showing the trials with the correct versus incorrect responses is: Were the participants really trying to complete the task or did they give up on the task because it was too difficult? If the latter were the case, one would expect an early decline of the MPD. But the opposite is true, instead. A small increase of the MPD was measured, suggesting that the participants were trying hard to complete the task until the time was up.

## Self-reported workload (NASA-TLX)

According to the results of the NASA-TLX questionnaire, the classification of the arithmetic tasks was done properly, since a statistically significant difference was found in the subjective mental workload across all three levels. The large contrast between the subjective mental and physical workload underlines that the task was predominantly mentally rather than physically demanding. Not to be overlooked are the roles of the subjective temporal demand and frustration. Looking at the increase of the MPD of the incorrect responses after 12 s for Level 3 (Fig. 9), it is plausible that, although the results were not statistically significant, this increase was caused by the time pressure of the task or the anxiety or frustration of not having solved the multiplication yet, instead of increased task demands.

## Blink rate

The relation between mental workload and blink rate has been unclear in the literature (e.g., *Kramer, 1990*; *Marquart, Cabrall & De Winter, 2015*; *Recarte et al., 2008*). The results in the present study show that the MBR was slightly higher for Level 3 than for Levels 1 and 2. Contrastingly, the MBR was higher during the low mental demand period (0–6.5 s) than during the high demand period (6.5–17.5). The temporal analysis (Fig. 10) indicated that people blinked particularly at those moments when the visual demand was reduced, such as right after the start of the task and right after the presentation of the multiplier. In summary, consistent with prior research, the relationship between mental workload

and blink rate is complex, and it appears that blink rate is governed not only by mental demands, but also by visual demands (see also *Marquart, Cabrall & De Winter, 2015*).

## Correlations between MPDC, NASA-TLX, and proportion of incorrect responses

Moderate to strong correlations were found between the MPDC and the proportion of incorrect responses. A similar but weaker effect was obtained between the MPDC and the NASA-TLX. Thus, the MPDC was higher for participants who gave more incorrect responses and who reported a higher workload in the NASA-TLX. Negative correlations between the pupil diameter and the proportion of correct responses were also found by *Ahern (1978)*, *Payne, Parry & Harasymiw (1968)* and *Recarte et al. (2008)*. These findings could be useful for determining the feasibility of using the pupil diameter in human-machine applications such as adaptive automation, which is "an approach to automation design where tasks are dynamically allocated between the human operator and computer systems" (*Byrne & Parasuraman, 1996*, p. 249).

## Conclusions and recommendations

It is concluded that the results of *Ahern (1978)* and *Klingner (2010)* have been accurately replicated with the SmartEye DR120 remote eye tracker. The Cohen $d_z$ effect size between the MPDC of Level 1 and Level 3 was 1.95 at maximum (at Point 8), which was about the same ($d_z = 1.91$) as for the NASA-TLX overall score. This finding demonstrates that pupil diameter measurements can be just as valid as the NASA-TLX. In our research, an attempt was made to provide more insight into the individual differences of TEPRs by means of a correlation analysis. Results showed a few moderate to strong correlations at the beginning of the calculation period between the MPDC and the NASA-TLX, on the one hand, and the percentage of incorrect responses, on the other.

Thus, it seems possible to assess workload by tracking the pupil diameter. However, the validity of pupil diameter measurements may need improvement before it could be implemented in practice. Future research could focus on improving signal analysis techniques that filter out effects other than mental workload, such as the light reflex. It is challenging to enhance the applicability of pupillometry towards tasks that require fixation on different types of targets. *Janisse (1977)* previously concluded that research that uses pictorial stimuli should "be interpreted with caution, and perhaps be discounted." (p. 77). One possible way to use the pupil diameter in visually complex tasks might be to correct in real time for the amount of light that enters the eye. Janisse proposed such approach as early as 1977: "The simultaneous monitoring of pupil size and eye movements (points of focus) as subjects view pictorial stimuli might allow one to mathematically 'correct' pupil size as a function of the brightness of the point on which the subject's gaze is falling at a given time." (p. 169). Because modern remote eye trackers measure gaze direction and pupil diameter simultaneously, such approach becomes within practical reach, as also discussed by *Klingner (2010)*. For further reading into approaches of pupillometry in complex visual environments, see *Palinko & Kun* (*2011*; a driving simulator), and *Klingner* (*2010*; visual search and map reading).

Additionally, validity could be improved by combining pupillometry with other physiological measures (e.g., *Haapalainen et al., 2010*; *Just, Carpenter & Miyake, 2003*; *Kahneman et al., 1969*; *Satterthwaite et al., 2007*; *Van der Molen et al., 1989*). For example, *Haapalainen et al. (2010)* used an electrocardiogram (ECG)-enabled armband, a remote eye tracker, and a wireless electroencephalogram (EEG) headset, to collect various physiological signals simultaneously. The authors concluded that the heat flux and heart rate variability in combination provided a classification accuracy of over 80% between conditions of low and high mental workload. In this study, the pupil diameter did not perform strongly as a classifier (57%), presumably due to data loss of the eye tracker. A primary advantage of pupillometry in such multivariate applications is that the pupil diameter reacts rapidly to changes in task conditions (cf. Fig. 5), while measures such as heat flux, galvanic skin response, or heart rate have considerably longer time constants.

### Funding
The authors received no funding for this work.

### Competing Interests
The authors declare there are no competing interests.

### Author Contributions
- Gerhard Marquart conceived and designed the experiments, performed the experiments, analyzed the data, contributed reagents/materials/analysis tools, wrote the paper, prepared figures and/or tables, performed the computation work, reviewed drafts of the paper.
- Joost de Winter conceived and designed the experiments, analyzed the data, contributed reagents/materials/analysis tools, wrote the paper, prepared figures and/or tables, performed the computation work, reviewed drafts of the paper.

### Ethics
The following information was supplied relating to ethical approvals (i.e., approving body and any reference numbers):

The research was approved by the Human Research Ethics Committee (HREC) of the Delft University of Technology (TU Delft). ('Workload Assessment for Mental Arithmetic Tasks using the Task-Evoked Pupillary Response: January 29, 2015).

### Data Availability
The following information was supplied regarding the deposition of related data:

The Experimenter software and analysis scripts are available as Supplemental Files but as the raw data files are quite large, they are currently hosted at: http://repository.tudelft.nl/assets/uuid:c34edcab-2734-4cd9-b060-67371eb3bab0/Supplementary_Material_Gerhard_Marquart.zip.

## Supplemental Information

Supplemental information for this article can be found online at http://dx.doi.org/10.7717/peerj-cs.16#supplemental-information.

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
