# Peer review of "Workload assessment for mental arithmetic tasks using the task-evoked pupillary response"

_PeerJ Computer Science, doi:10.7717/peerj-cs.16_

## Round 0.1 · original submission · Minor Revisions

· Academic Editor

Minor Revisions

We all agree this is a solid and interesting pice of research and well worthy of publication, albeit with some minor revisions. The two reviewers have outlined a number of improvements to the paper, which I ask the authors to address.

·

Basic reporting

The paper is very clearly, if rather densely written. The use of acronyms does require a certain mental agility from the reader but I am not sure there is a way round this and the authors do reiterate the basic terms in the Results section.

There is a good introduction to motivate the work, primarily a replication of the previous work of Ahern and Klingner with better temporal resolution and only visual stimuli. It is not entirely clear from the way the paper is written that the NASA TLX is a new measure added to this study. See for instance the abstract. From reading Klingner's thesis, I see that the workload measures are an introduction for this study but this should be addressed with more clear phrasing in the paper itself. I think there might be room to go more into the details of exactly what Ahern and Klingner did so that the contrast with the current work is clearer. In particular, many measures are used: MPD, MPDC, MPDCR. Is there a reason for this? I would have thought only MPDC was needed. Unless of course this is part of the replication (I can't see Ahern so can't be sure). So some clarity there would help.

I also felt that the Results section could be more coherently organised. Understandably there are lots of results and anything that helps the reader navigate them would be useful. For instance, MPDC effect sizes are reported several paragraphs after the MPDC is analysed. It makes logical sense (and is also analytically appropriate) to put these together. Similarly, NASA TLX analysis is separated from correlations with the NASA TLX measures. I found myself skipping around between tables, figures and sections of the Results and I think some of the skipping could have been removed.

Experimental design

The experimental design seems to be very appropriate and the experiment seems to have been conducted very well.

Validity of the findings

The findings seem good to me though I would query a couple of points. I was not aware that Tukey's HSD was appropriate for repeated measures ANOVA. I finally tracked down an online resource by Howell, of statistical textbook fame so a decent source if not an authoritative outlet. He reckons that such follow ups are not sensitive to violations of assumption and recommends Bonferroni corrected t-tests. The authors may have a more authoritative reason for their approach but I feel it should be referenced as this is not common practice (unlike most of the other aspects of the statistical testing). At the same time, the use of Bonferroni corrections can be very conservative. There is room to discuss this.

I am very happy that this is replication work with the added in NASA TLX measures. Equipment does change and provides new opportunities. And of course any measurement instrument operationalises the measures in new ways. It is good to have up to date support for commonly accepted historical results.

·

Basic reporting

No Comments

Experimental design

The Design of the experiment was fine for a repeated measures study - counterbalancing was in place.

I would have liked more details about lighting levels in the testing room. How could the authors be sure that lighting was consistent across all testing conditions? Was lighting level measured?

Validity of the findings

The results of the experiment are rigorously reported and conclusions are accurate and clearly stated.
- Given that the authors measured 3 different aspects of pupil (MPD, MPDC, MPDCR), I would prefer to see some clear recommendations about strengths and weaknesses of each variable, especially for HCI evaluation. Is one measure less sensitive than others to fluctuations in naturalistic light.
- It is good that MPDC is as sensitive to subjective workload but could authors spell out advantage of physiological monitoring for the naive reader in the conclusions?
- The presentation of the eye blink data in Fig 10 was hard to understand, more text required to explain this aspect
- the task used by the authors had minimal visual demand but high cognitive demand, could they please speculate in the discussion section about how their results may generalise to other types of tasks, especially ones where visual workload may be manipulated, e.g. locating information on different screen designs or web pages.

Comments for the author

I appreciate the thorough reporting of data, but I felt that Figure 5 was not really necessary

---

## Round 0.2 · accepted · Accept

· Academic Editor

Accept

Thank your for your careful consideration of our comments and the appropriate adjustments which you made to the manuscript. They are all very satisfactory.